# Learning Algebraic Representation for Abstract Spatial-Temporal Reasoning

## Abstract

Is intelligence realized by connectionist or classicist? While connectionist approaches have achieved superhuman performance, there has been growing evidence that such task-specific superiority is particularly fragile in *systematic generalization*. This observation lies in the central debate (Fodor et al., 1988; Fodor & McLaughlin, 1990) between connectionist and classicist, wherein the latter continually advocates an *algebraic* treatment in cognitive architectures. In this work, we follow the classicist's call and propose a hybrid approach to improve systematic generalization in reasoning. Specifically, we showcase a prototype with algebraic representations for the abstract spatial-temporal reasoning task of Raven's Progressive Matrices (RPM) and present the ALgebra-Aware Neuro-Semi-Symbolic (ALANS$^2$) learner. The ALANS$^2$ learner is motivated by abstract algebra and the representation theory. It consists of a neural visual perception frontend and an algebraic abstract reasoning backend: the frontend summarizes the visual information from object-based representations, while the backend transforms it into an algebraic structure and induces the hidden operator on-the-fly. The induced operator is later executed to predict the answer's representation, and the choice most similar to the prediction is selected as the solution. Extensive experiments show that by incorporating an algebraic treatment, the ALANS$^2$ learner outperforms various pure connectionist models in domains requiring systematic generalization. We further show that the algebraic representation learned can be decoded by isomorphism and used to *generate* an answer.

## 1 Introduction

> "Thought is in fact a kind of Algebra."                —William James (James, 1891)

Imagine you are given two alphabetical sequences of "$c, b, a$" and "$d, c, b$", and asked to fill in the missing element in "$e, d, ?$". In nearly no time will one realize the answer to be $c$. However, more surprising for human learning is that, effortlessly and instantaneously, we can "freely generalize" (Marcus, 2001) the solution to any partial consecutive ordered sequences. While believed to be innate in early development for human infants (Marcus et al., 1999), such systematic generalizability has constantly been missing and proven to be particularly challenging in existing connectionist models (Lake & Baroni, 2018; Bahdanau et al., 2019). In fact, such an ability to entertain a given thought and semantically related contents strongly implies an abstract algebra-like treatment (Fodor et al., 1988); in literature, it is referred to as the "language of thought" (Fodor, 1975), "physical symbol system" (Newell, 1980), and "algebraic mind" (Marcus, 2001). However, in stark contrast, existing connectionist models tend only to capture statistical correlation (Lake & Baroni, 2018; Kansky et al., 2017; Chollet, 2019), rather than providing any account for a structural inductive bias where systematic algebra can be carried out to facilitate generalization.

This contrast instinctively raises a question—what constitutes such an *algebraic* inductive bias? We argue that the foundation of the modeling counterpart to the algebraic treatment in early human development (Marcus, 2001; Marcus et al., 1999) lies in algebraic computations set up on mathematical axioms, a form of formalized human intuition and the starting point of modern mathematical reasoning (Heath et al., 1956; Maddy, 1988). Of particular importance to the basic building blocks of algebra is the Peano Axiom (Peano, 1889). In the Peano Axiom, the essential components of algebra, the algebraic set and corresponding operators over it, are governed by three statements: (1) the existence of at least one element in the field to study ("zero" element), (2) a successor function that is recursively applied to all elements and can, therefore, span the entire field, and (3) the principle of

mathematical induction. Building on such a fundamental axiom, we begin to form the notion of an algebraic set and induce the operator along with it to construct an algebraic structure. We hypothesize that such a treatment of algebraic computations set up on fundamental axioms is essential for a model's systematic generalizability, the lack of which will only make it sub-optimal.

To demonstrate the benefits of such an algebraic treatment in systematic generalization, we showcase a prototype for Raven's Progressive Matrices (RPM) (Raven, 1936; Raven & Court, 1998), an exemplar task for abstract spatial-temporal reasoning (Santoro et al., 2018; Zhang et al., 2019a). In this task, an agent is given an incomplete $3 \times 3$ matrix consisting of eight context panels with the last one missing, and asked to pick one answer from a set of eight choices that best completes the matrix. Human's reasoning capability of solving this abstract reasoning task has been commonly regarded as an indicator of "general intelligence" (Carpenter et al., 1990) and "fluid intelligence" (Spearman, 1923; 1927; Hofstadter, 1995; Jaeggi et al., 2008). In spite of the task being one that ideally requires abstraction, algebraization, induction, and generalization (Raven, 1936; Raven & Court, 1998; Carpenter et al., 1990), recent endeavors unanimously propose pure connectionist models that attempt to circumvent such intrinsic cognitive requirements (Santoro et al., 2018; Zhang et al., 2019a;b; Wang et al., 2020; Zheng et al., 2019; Hu et al., 2020; Wu et al., 2020). However, these methods' inefficiency is also evident in systematic generalization; they struggle to extrapolate to domains beyond training, as pointed out in (Santoro et al., 2018; Zhang et al., 2019b) and shown later in this paper.

To address the issue, we introduce the ALgebra-Aware Neuro-Semi-Symbolic (ALANS$^2$) learner. At a high-level, the ALANS$^2$ learner is embedded in a general neuro-symbolic architecture (Yi et al., 2018; Mao et al., 2019; Han et al., 2019; Yi et al., 2020) but has on-the-fly operator learnability and hence semi-symbolic. Specifically, it consists of a neural visual perception frontend and an algebraic abstract reasoning backend. For each RPM instance, the neural visual perception frontend first slides a window over each panel to obtain the object-based representations (Kansky et al., 2017; Wu et al., 2017) for every object. A belief inference engine latter aggregates all object-based representations in each panel to produce the probabilistic *belief state*. The algebraic abstract reasoning backend then takes the belief states of the eight context panels, treats them as snapshots on an algebraic structure, lifts them into a matrix-based algebraic representation built on the Peano Axiom and the representation theory (Humphreys, 2012), and induces the hidden operator in the algebraic structure by solving an inner optimization (Colson et al., 2007; Bard, 2013). The algebraic representation for the answer is predicted by executing the induced operator: its corresponding set element is decoded by isomorphism established in the representation theory, and the final answer is selected as the one most similar to the prediction.

The ALANS$^2$ learner enjoys several benefits in abstract reasoning with an algebraic treatment:

1. Unlike previous monolithic models, the ALANS$^2$ learner offers a more interpretable account of the entire abstract reasoning process: the neural visual perception frontend extracts object-based representations and produces belief states of panels by explicit probability inference, whereas the algebraic abstract reasoning backend induces the hidden operator in the algebraic structure. The corresponding representation for the final answer is obtained by executing the induced operator, and the choice panel with minimum distance is selected. This process much resembles the top-down bottom-up strategy in human reasoning: humans reason by inducing the hidden relation, executing it to generate a feasible solution in mind, and choosing the most similar answer available (Carpenter et al., 1990). Such a strategy is missing in recent literature (Santoro et al., 2018; Zhang et al., 2019a;b; Wang et al., 2020; Zheng et al., 2019; Hu et al., 2020; Wu et al., 2020).
2. While keeping the semantic interpretability and end-to-end trainability in existing neuro-symbolic frameworks (Yi et al., 2018; Mao et al., 2019; Han et al., 2019; Yi et al., 2020), ALANS$^2$ is what we call semi-symbolic in the sense that the symbolic operator can be learned and concluded on-the-fly without manual definition for every one of them. Such an inductive ability also enables a greater extent of the desired generalizability.
3. By decoding the predicted representation in the algebraic structure, we can also generate an answer that satisfies the hidden relation in the context.

This work makes three major contributions: (1) We propose the ALANS$^2$ learner. Compared to existing monolithic models, the ALANS$^2$ learner adopts a neuro-semi-symbolic design, where the problem-solving process is decomposed into neural visual perception and algebraic abstract reasoning. (2) To demonstrate the efficacy of incorporating an algebraic treatment in abstract spatial-temporal reasoning, we show the superior systematic generalization ability of the proposed ALANS$^2$ learner in various extrapolatory RPM domains. (3) We present analyses into both neural visual perception and algebraic abstract reasoning. We also show the generative potential of ALANS$^2$.

## 2    RELATED WORK

**Quest for Symbolized Manipulation**    The idea to treat thinking as a mental language can be dated back to Augustine (Augustine, 1876; Wittgenstein, 1953). Since the 1970s, this school of thought has undergone a dramatic revival as the quest for a symbolized manipulation in cognitive modeling, such as "language of thought" (Fodor, 1975), "physical symbol system" (Newell, 1980), and "algebraic mind" (Marcus, 2001). In their study, connectionist's task-specific superiority and inability to generalize beyond training (Kansky et al., 2017; Chollet, 2019; Santoro et al., 2018; Zhang et al., 2019a) have been hypothetically linked to a lack of such symbolized algebraic manipulation (Lake & Baroni, 2018; Chollet, 2019; Marcus, 2020). With evidence that an algebraic treatment adopted in early human development (Marcus et al., 1999) can potentially address the issue (Bahdanau et al., 2019; Mao et al., 2019; Marcus, 2020), classicist (Fodor et al., 1988) approaches for generalizable reasoning used in programs (McCarthy, 1960) and blocks world (Winograd, 1971) have resurrected. As a hybrid approach to bridge connectionist and classicist, recent developments lead to neuro-symbolic architectures. In particular, Yi et al. (2018) demonstrate a neuro-symbolic prototype for visual question answering, where a perception module and a language parsing module are separately trained, and the predefined logic operators associated with language tokens are chained to process the visual information. Mao et al. (2019) soften the predefined operators to afford end-to-end training with only question answers. Han et al. (2019) and Yi et al. (2020) use the hybrid architecture for metaconcept learning and temporal causal learning, respectively. ALANS$^2$ follows the classicist's call but adopts a neuro-*semi*-symbolic architecture: it is end-to-end trainable as opposed to Yi et al. (2018; 2020) and the operator can be learned and concluded on-the-fly without manual specification (Yi et al., 2018; Mao et al., 2019; Han et al., 2019; Yi et al., 2020).

**Abstract Visual Reasoning**    Recent works by Santoro et al. (2018) and Zhang et al. (2019a) arouse the community's interest in abstract visual reasoning, where the task of Raven's Progressive Matrices (RPM) is introduced as such a measure for intelligent agents. Initially proposed as an intelligence quotient test for humans (Raven, 1936; Raven & Court, 1998), RPM is believed to be strongly correlated with human's general intelligence (Carpenter et al., 1990) and fluid intelligence (Spearman, 1923; 1927; Hofstadter, 1995; Jaeggi et al., 2008). Early RPM-solving systems employ symbolic representations based on hand-designed features and assume access to the underlying logics (Carpenter et al., 1990; Lovett et al., 2009; 2010; Lovett & Forbus, 2017). Another stream of research on RPM recruits similarity-based metrics to select the most similar answer from the choices (Little et al., 2012; McGreggor & Goel, 2014; McGreggor et al., 2014; Mekik et al., 2018; Shegheva & Goel, 2018). However, their hand-defined visual features are unable to handle uncertainty from imperfect perception, and directly assuming access to the logic operations simplifies the problem. Recently proposed data-driven approaches arise from the availability of large datasets: Santoro et al. (2018) extend a pedagogical RPM generation method (Wang & Su, 2015), whereas Zhang et al. (2019a) use a stochastic image grammar (Zhu et al., 2007) and introduce structural annotations in it, which Hu et al. (2020) further refine to avoid shortcut solutions by statistics in candidate panels. Despite the fact that RPM intrinsically requires one to perform abstraction, algebraization, induction, and generalization, existing methods bypass such cognitive requirements using a single feedforward pass in connectionist models: Santoro et al. (2018) use a relational module (Santoro et al., 2017), Steenbrugge et al. (2018) augment it with a VAE (Kingma & Welling, 2013), Zhang et al. (2019a) assemble a dynamic tree, Hill et al. (2019) arrange the data in a contrasting manner, Zhang et al. (2019b) propose a contrast module, Zheng et al. (2019) formulate it in a student-teacher setting, Wang et al. (2020) build a multiplex graph network, Hu et al. (2020) aggregate features from a hierarchical decomposition, and Wu et al. (2020) apply a scattering transformation to learn objects, attributes, and relations. In contrast, ALANS$^2$ attempts to fulfill the cognitive requirements in a neuro-semi-symbolic framework: the perception frontend abstracts out visual information, and the reasoning backend induces the hidden operator in an algebraic structure.

## 3    THE ALANS$^2$ LEARNER

In this section, we introduce the ALANS$^2$ learner for the RPM problem. In each RPM instance, an agent is given an incomplete $3 \times 3$ panel matrix with the last entry missing and asked to induce the operator hidden in the matrix and choose from eight choice panels one that follows it. Formally, let the answer variable be denoted as $y$, the context panels as $\{I_{o,i}\}_{i=1}^8$, and choice panels as $\{I_{c,i}\}_{i=1}^8$. Then the problem can be formulated as estimating $P(y \mid \{I_{o,i}\}_{i=1}^8, \{I_{c,i}\}_{i=1}^8)$. According to the common design (Santoro et al., 2018; Zhang et al., 2019a; Carpenter et al., 1990), there is one operator that governs each panel attribute. Hence, by assuming independence among attributes, we

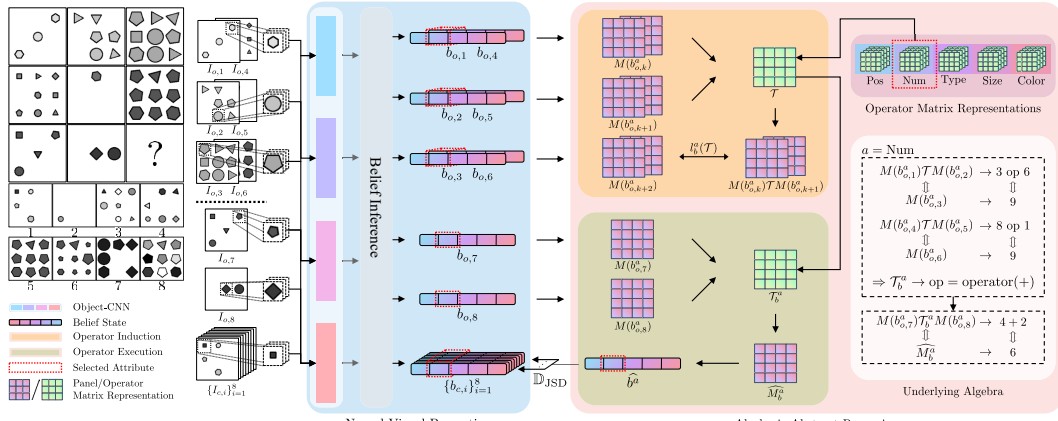

Figure 1: An overview of the ALANS$^2$ learner. For an RPM instance, the neural visual perception module produces the belief states for all panels: an object CNN extracts object attribute distributions for each image region, and a belief inference engine marginalizes them out to obtain panel attribute distributions. For each panel attribute, the algebraic abstract reasoning module transforms the belief states into matrix-based algebraic representations and induces hidden operators by solving inner optimizations. The answer representations are obtained by executing the induced operators, and the choice most similar to the prediction is selected as the solution. An example of the underlying discrete algebra and its correspondence is also shown on the right.

propose to factorize the probability as

$$P(y = n \mid \{I_{o,i}\}_{i=1}^8, \{I_{c,i}\}_{i=1}^8) \propto \prod_a \sum_{\mathcal{T}^a} P(y^a = n \mid \mathcal{T}^a, \{I_{o,i}\}_{i=1}^8, \{I_{c,i}\}_{i=1}^8) P(\mathcal{T}^a \mid \{I_{o,i}\}_{i=1}^8), \quad (1)$$

where $y^a$ denotes the answer selection based only on attribute $a$ and $\mathcal{T}^a$ the operator on $a$.

**Overview**   As shown in Fig. 1, the ALANS$^2$ learner decomposes the process into perception and reasoning: the neural visual perception frontend extracts the *belief states* from each of the sixteen panels, whereas the algebraic abstract reasoning backend views an instance as an example in an abstract algebra structure, transforms belief states into *algebraic representations* by representation theory, *induces* the hidden operators, and *executes* the operators to predict the representation of the answer. Therefore, in Eq. (1), the operator distribution is modeled by the fitness of an operator and the answer distribution by the distance between the predicted representation and that of a candidate.

### 3.1 Neural Visual Perception

The neural visual perception frontend consists of an object CNN and a belief inference engine. It is responsible for extracting the belief states for each of the sixteen (context and choice) panels.

**Object CNN**   For each panel, we use a sliding window to traverse the spatial domain of the image and feed each image region into an object CNN. The CNN has four branches, producing for each region its object attribute distributions, including objectiveness (if the region contains an object), type, size, and color. Distributions of type, size, and color are conditioned on an object's existence.

**Belief Inference Engine**   The belief inference engine summarizes the panel attribute distributions (over position, number, type, size, and color) by marginalizing out all object attribute distributions (over objectiveness, type, size, and color). As an example, the distribution of the panel attribute of Number can be computed as such: for $N$ image regions and their predicted objectiveness

$$P(\text{Number} = k) = \sum_{R^o} \prod_{j=1}^N P(r_j^o = R_j^o), \quad (2)$$

where $P(r_j^o)$ denotes the $j$th region's estimated objectiveness distribution, and $R^o$ is a binary sequence of length $N$ that sums to $k$. All panel attribute distributions compose the *belief state* of a panel. In the following, we denote the belief state as $b$ and the distribution of an attribute $a$ as $P(b^a)$.

### 3.2 Algebraic Abstract Reasoning

Given the belief states of both context and choice panels, the algebraic abstract reasoning backend concerns the induction of hidden operators and the prediction of answer representations for each

attribute. The fitness of induced operators is used for estimating the operator distribution and the difference between the prediction and the choice panel for estimating the answer distribution.

**Algebraic Underpinning**  Without loss of generality, here we assume row-wise operators. For each attribute, under perfect perception, the first two rows in an RPM instance provide snapshots into an example of *magma* (Hausmann & Ore, 1937) constrained to an integer-indexed set, the simplest group-like algebra structure that is closed under a binary operator. To see this, note that an accurate perception module would see each panel attribute as a deterministic set element. Therefore, RPM instances with unary operators, such as progression, are magma examples with special binary operators where one operand is constant. Instances with binary operators, such as arithmetics, directly follow the magma properties. Those with ternary operators are ones with unary operators on a three-tuple set defined on rows.

**Algebraic Representation**  A systematic algebraic view allows us to felicitously recruit ideas in representation theory (Humphreys, 2012) to glean the hidden properties in the abstract structures: it makes abstract algebra amenable by reducing it onto linear algebra. Following the same spirit, we propose to lift both the set elements and the hidden operators to a learnable matrix space. To encode the set element, we employ the Peano Axiom (Peano, 1889). According to the Peano Axiom, an integer-indexed set can be constructed by (1) a zero element ($\mathbf{0}$), (2) a successor function ($S(\cdot)$), and (3) the principle of mathematical induction, such that the $k$th element is

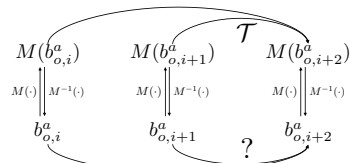

Figure 2: Isomorphism between the abstract algebra and the matrix-based representation. Operator induction reduced to matrices.

encoded as $S^k(\mathbf{0})$. Specifically, we instantiate the zero element as a learnable matrix $M_0$ and the successor function as the matrix-matrix product parameterized by $M$. In an attribute-specific manner, the representation of an attribute taking the $k$th value is $(M^a)^k M_0^a$. For operators, we consider them to live in a learnable matrix group of a corresponding dimension, such that the action of an operator on a set can be represented as matrix multiplication. Such algebraic representations establish an isomorphism between the matrix space and the abstract algebraic structure: abstract elements on the algebraic structure have a bijective mapping to/from the matrix space, and inducing the abstract relation can be reduced to solving for a matrix operator. See Fig. 2 for a graphical illustration of the isomorphism.

**Operator Induction**  Operator induction concerns about finding a concrete operator in the abstract algebraic structure. By the property of closure, we formulate it as an inner-level regularized linear regression problem: a binary operator $\mathcal{T}_b^a$ in a magma example for attribute $a$ minimizes

$$\arg\min_{\mathcal{T}} \ell_b^a(\mathcal{T}) = \sum_i \mathbb{E}\left[\|M(b_{o,i}^a)\mathcal{T}M(b_{o,i+1}^a) - M(b_{o,i+2}^a)\|_F^2\right] + \lambda_b^a\|\mathcal{T}\|_F^2, \quad (3)$$

where under visual uncertainty, we take the expectation with respect to the distributions in the belief states of context panels $P(b_{o,i}^a)$ in the first two rows, and denote its algebraic representation as $M(b_{o,i}^a)$. For unary operators, one operand can be treated as constant and absorbed into $\mathcal{T}$. Note that Eq. (3) admits a closed-form solution (see Appendix for details). Therefore, the operator can be learned and adapted for different instances of binary relations and concluded on-the-fly. Such a design also simplifies the recent neuro-symbolic approaches, where every single symbol operator needs to be hand-defined (Yi et al., 2018; Mao et al., 2019; Han et al., 2019; Yi et al., 2020). Instead, we only specify an inner-level optimization framework and allow symbolic operators to be quickly induced based on the neural observations, while keeping the semantic interpretability in the neuro-symbolic methods. Therefore, we term such a design semi-symbolic.

The operator probability in Eq. (1) is then modeled by each operator type's fitness, *e.g.*, for binary,

$$P(\mathcal{T}^a = \mathcal{T}_b^a \mid \{I_{o,i}\}_{i=1}^8) \propto \exp(-\ell_b^a(\mathcal{T}_b^a)). \quad (4)$$

**Operator Execution**  To predict the algebraic representation of the answer, we solve another inner-level optimization similar to Eq. (3), but now treating the representation of the answer as a variable:

$$\widehat{M_b^a} = \arg\min_M \ell_b^a(M) = \mathbb{E}[\|M(b_{o,7}^a)\mathcal{T}_b^a M(b_{o,8}^a) - M\|_F^2], \quad (5)$$

where the expectation is taken with respect to context panels in the last row. The optimization also admits a closed-form solution (see Appendix for details), which corresponds to the execution of the induced operator in Eq. (3).

The predicted representation is decoded probabilistically as the predicted belief state of the solution,

$$P(\widehat{b^a} = k \mid \mathcal{T}^a) \propto \exp(-\|\widehat{M^a} - (M^a)^k M_0^a\|_F^2). \tag{6}$$

**Answer Selection**    Based on Eqs. (1) and (4), estimating the answer distribution is now boiled down to estimating the conditional answer distributions for each attribute. Here, we propose to model it based on the Jensen–Shannon Divergence (JSD) of the predicted belief state and that of a choice,

$$P(y^a = n \mid \mathcal{T}^a, \{I_{o,i}\}_{i=1}^8, \{I_{c,i}\}_{i=1}^8) \propto \exp(-\mathbb{D}_{\text{JSD}}(P(\widehat{b^a} \mid \mathcal{T}^a)\|P(b_{c,n}^a))). \tag{7}$$

**Discussion**    The algebraic abstract reasoning module offers a computational and interpretable counterpart to human-like reasoning in RPM (Carpenter et al., 1990). Specifically, the induction component resembles the fluid intelligence, where one quickly induces the hidden operator by observing the context panels. The execution component synthesizes an image by executing the induced operator, and the choice most similar to the image is selected as the answer. We also note that by decoding the predicted representation in Eq. (6), a solution can be *generated*: by sequentially selecting the most probable operator and the most probable attribute value, a rendering engine can directly render the solution. The reasoning backend also enables end-to-end training: by integrating the belief states from neural perception, the module conducts both induction and execution in a soft manner, such that the gradients can be back-propagated and the learner jointly trained.

### 3.3    LEARNING OBJECTIVE

We train the entire ALANS$^2$ learner by minimizing the cross-entropy loss between the estimated answer distribution and the ground-truth selection, *i.e.*,

$$\min_{\theta, \{M_0^a\}, \{M^a\}} \ell(P(y \mid \{I_{o,i}\}_{i=1}^8, \{I_{c,i}\}_{i=1}^8), y_\star), \tag{8}$$

where $\ell(\cdot)$ denotes the cross-entropy loss, $y_\star$ the ground-truth selection, $\theta$ the parameters in the object CNN, and $\{M_0^a\}$ and $\{M^a\}$ the zero elements and the successor functions for element encodings, respectively. Note notations are simplified by making the dependency on parameters implicit.

However, we notice in practice that with only the cross-entropy loss on the ground-truth selection, the ALANS$^2$ learner experiences difficulty in convergence. Without a proper guidance, the object CNN does not produce meaningful object-based representations. Therefore, following the discussion in (Santoro et al., 2018; Zhang et al., 2019a; Wang et al., 2020), we augment training with an auxiliary loss on the distribution of the operator, *i.e.*,

$$\min_{\theta, \{M_0^a\}, \{M^a\}} \ell(P(y \mid \{I_{o,i}\}_{i=1}^8, \{I_{c,i}\}_{i=1}^8), y_\star) + \sum_a \lambda^a \ell(P(\mathcal{T}^a \mid \{I_{o,i}\}_{i=1}^8), y_\star^a), \tag{9}$$

where $y_\star^a$ denotes the ground-truth operator selection for attribute $a$, and $\lambda^a$ balances the trade-off.

## 4    EXPERIMENTS

A cognitive architecture with systematic generalization is believed to demonstrate the following three principles (Fodor et al., 1988; Marcus, 2001; 2020): (1) systematicity, (2) productivity, and (3) localism. Systematicity requires an architecture to be able to entertain "semantically related" contents after understanding a given thought. Productivity states that the awareness of a constituent implies that of a recursive application of the constituent, and vice versa for localism.

To verify the effectiveness of an algebraic treatment in systematic generalization, we showcase the superiority of the proposed ALANS$^2$ learner on the three principles in the abstract spatial-temporal reasoning task of RPM. Specifically, we use the generation methods proposed in Zhang et al. (2019a) and Hu et al. (2020) to generate RPM problems and carefully split training and testing to construct the three regimes. The former generates candidates by perturbing only one attribute of the correct answer while the later modifies attribute values in a hierarchical manner to avoid shortcut solutions by pure statistics. Both methods categorize relations in RPM into three types, according to Carpenter et al. (1990): unary (Constant and Progression), binary (Arithmetic), and ternary (Distribution of Three), each of which comes with several instances. Grounding the principles into learning abstract relations in RPM, we fix the configuration to be $3 \times 3$Grid and generate the following data splits for evaluation (see Appendix for details):

- Systematicity: the training set contains only a subset of instances for each type of relation, while the test set all other relation instances.

- Productivity: as the binary relation results from a recursive application of the unary relation, the training set contains only unary relations, whereas the test set only binary relations.
- Localism: the training and testing sets in the productivity split are swapped to study localism.

We follow Zhang et al. (2019a) to generate $10,000$ instances for each split and assign 6 folds for training, 2 folds for validation, and 2 folds for testing.

**Experimental Setup**  We evaluate the systematic generalizability of the proposed ALANS$^2$ learner on the above three splits, and compare the ALANS$^2$ learner with other baselines, including ResNet, ResNet+DRT (Zhang et al., 2019a), WReN (Santoro et al., 2018), CoPINet (Zhang et al., 2019b), MXGNet (Wang et al., 2020), LEN (Zheng et al., 2019), HriNet (Hu et al., 2020), and SCL (Wu et al., 2020). We use either official or public implementations that reproduce the original results.

Table 1: Model performance on different aspects of systematic generalization. The performance is measured by accuracy and reported on the test sets. Upper: results on datasets generated by Zhang et al. (2019a). Lower: results on datasets generated by Hu et al. (2020).

| Method | MXGNet | ResNet+DRT | ResNet | HriNet | LEN | WReN | SCL | CoPINet | ALANS$^2$ | ALANS$^2$-Ind |
|---|---|---|---|---|---|---|---|---|---|---|
| Systematicity | 20.95% | 33.00% | 27.35% | 28.05% | 40.15% | 35.20% | 37.35% | 59.30% | **78.45%** | 52.70% |
| Productivity | 30.40% | 27.95% | 27.05% | 31.45% | 42.30% | 56.95% | 51.10% | 60.00% | **79.95%** | 36.45% |
| Localism | 28.80% | 24.90% | 23.05% | 29.70% | 39.65% | 38.70% | 47.75% | 60.10% | **80.50%** | 59.80% |
| Average | 26.72% | 28.62% | 25.82% | 29.73% | 40.70% | 43.62% | 45.40% | 59.80% | **79.63%** | 48.65% |
| Systematicity | 13.35% | 13.50% | 14.20% | 21.00% | 17.40% | 15.00% | 24.90% | 18.35% | **64.80%** | 52.80% |
| Productivity | 14.10% | 16.10% | 20.70% | 20.35% | 19.70% | 17.95% | 22.20% | 29.10% | **65.55%** | 32.10% |
| Localism | 15.80% | 13.85% | 17.45% | 24.60% | 20.15% | 19.70% | 29.95% | 31.85% | **65.90%** | 50.70% |
| Average | 14.42% | 14.48% | 17.45% | 21.98% | 19.08% | 17.55% | 25.68% | 26.43% | **65.42%** | 45.20% |

**Systematic Generalization**  Table 1 shows the performance of various models on systematic generalization, *i.e.*, systematicity, productivity, and localism. Compared to results reported in Santoro et al. (2018); Zhang et al. (2019a;b); Wang et al. (2020); Zheng et al. (2019); Hu et al. (2020); Wu et al. (2020), all pure connectionist models experience a devastating performance drop when it comes to the critical cognitive requirements on systematic generalization, indicating that pure connectionist models fail to perform abstraction, algebraization, induction, or generalization needed in solving the abstract reasoning task; instead, they seem to only take a shortcut to bypass them. In particular, MXGNet (Wang et al., 2020)'s superiority is diminishing in systematic generalization. Despite of learning with structural annotations, ResNet+DRT (Zhang et al., 2019a) does not fare better than its base model. The recently proposed HriNet (Hu et al., 2020) slightly improves on ResNet in this aspect, with LEN (Zheng et al., 2019) being only marginally better. WReN (Santoro et al., 2018), on the other hand, shows oscillating performance across the three regimes. Evaluated under systematic generation, SCL (Wu et al., 2020) and CoPINet (Zhang et al., 2019b) also far deviate from their "superior performance". These observations suggest that pure connectionist models highly likely learn from variation in visual appearance rather than the algebra underlying the problem.

Embedded in a neural-semi-symbolic framework, the proposed ALANS$^2$ learner improves on systematic generalization by a large margin. With an algebra-aware design, the model is considerably stable across different principles of systematic generalization. The algebraic representations learned in relations of either a constituent or a recursive composition naturally support productivity and localism, while semi-symbolic inner optimization further allows various instances of an operator type to be induced from the algebraic representations and boosts systematicity. The importance of the algebraic representations is made more significant in the ablation study: ALANS$^2$-Ind, with algebraic representation replaced by independent encodings and the algebraic isomorphism broken, shows inferior performance. The ALANS$^2$ learner also enables diagnostic tests into its jointly learned perception module and reasoning module, in contrast to the black-box-like connectionist counterparts.

**Analysis into Perception and Reasoning**  The neural-semi-symbolic design allows analyses into both perception and reasoning. To evaluate the neural perception module and the algebraic reasoning module, we extract region-based object attribute annotations from the dataset generation methods (Zhang et al., 2019a; Hu et al., 2020) and categorize all relations into three types, *i.e.*, unary, binary, and ternary, respectively.

Table 2 shows the perception module's performance on the test sets in the three regimes of systematic generalization. We note that in order for the ALANS$^2$ learner to achieve the desired results shown in Table 1, ALANS$^2$ learns to construct the concept of objectiveness perfectly. The model also shows a fairly accurate prediction accuracy on the attributes of type and size. However, on the texture-related concept of color, ALANS$^2$ fails to develop a reliable notion on it. Despite that, the general prediction accuracy of the perception module is still surprising, considering that the perception module is only

jointly learned with ground-truth annotations on answer selections. The relatively lower accuracy on color could be attributed to its larger space compared to other attributes.

Table 2: Perception accuracy of the proposed ALANS$^2$ learner, measured by whether the module can correctly predict an attribute's value. Left: results on datasets genereted by Zhang et al. (2019a). Right: results on datasets genereted by Hu et al. (2020).

| Object Attribute | Objectiveness | Type | Size | Color | Object Attribute | Objectiveness | Type | Size | Color |
|---|---|---|---|---|---|---|---|---|---|
| Systematicity | 100.00% | 99.95% | 94.65% | 71.35% | Systematicity | 100.00% | 96.34% | 92.36% | 63.98% |
| Productivity | 100.00% | 99.97% | 98.04% | 77.61% | Productivity | 100.00% | 94.28% | 97.00% | 69.89% |
| Localism | 100.00% | 95.65% | 98.56% | 80.05% | Localism | 100.00% | 95.80% | 98.36% | 60.35% |
| Average | 100.00% | 98.52% | 97.08% | 76.34% | Average | 100.00% | 95.47% | 95.91% | 64.74% |

Table 3: Reasoning accuracy of the proposed ALANS$^2$ learner, measured by whether the module can correctly predict the type of a relation on an attribute. Left: results on datasets genereted by Zhang et al. (2019a). Right: results on datasets generated by Hu et al. (2020).

| Relation on | Position | Number | Type | Size | Color | Relation on | Position | Number | Type | Size | Color |
|---|---|---|---|---|---|---|---|---|---|---|---|
| Systematicity | 72.04% | 82.14% | 81.50% | 80.80% | 40.40% | Systematicity | 69.96% | 80.34% | 83.50% | 80.85% | 28.85% |
| Productivity | - | 98.75% | 89.50% | 72.10% | 33.95% | Productivity | - | 99.10% | 87.95% | 68.50% | 23.10% |
| Localism | - | 74.70% | 44.25% | 56.40% | 54.20% | Localism | - | 70.55% | 36.65% | 42.30% | 33.20% |
| Average | 72.04% | 85.20% | 71.75% | 69.77% | 42.85% | Average | 69.96% | 83.33% | 69.37% | 63.88% | 28.38% |

Table 3 lists the reasoning module's performance during testing for the three aspects. Note that on position, the unary operator (shifting) and binary operator (set arithemtics) do not systematically imply each other. Hence, we do not count them as probes into productivity and localism. In general, we notice that the better the perception accuracy on one attribute, the better the performance on reasoning. However, we also note that despite the relatively accurate perception of objectiveness, type, and size, near perfect reasoning is never guaranteed. This deficiency is due to the perception uncertainty handled by expectation in Eq. (3): in spite of correctness when we take $\arg\max$, marginalizing by expectation will unavoidably introduce noise into the reasoning process. Therefore, an ideal reasoning module requires the perception frontend to be not only correct but also certain. Computationally, one can sample from the perception module and optimize Eq. (9) using REINFORCE (Williams, 1992). However, the credit assignment problem and variance in gradient estimation will further complicate training.

**Generative Potential** Compared to existing discriminative-only RPM-solving methods, the proposed ALANS$^2$ learner is unique in its generative potential. As mentioned above, the final panel attribute can be decoded by sequentially selecting the most probable hidden operator and the attribute value. When equipped with a rendering engine, a solution can be generated. Here, we use the rendering program released by Zhang et al. (2019a) to demonstrate such a generative potential in the proposed ALANS$^2$ learner. Fig. 3 shows examples where the solutions are generated by ALANS$^2$. Such a generative ability is a computational counterpart to human reasoning: ALANS$^2$ *selects* the one most similar to a *synthesized image* from the pool of candidates, which resembles human's top-down bottom-up reasoning.

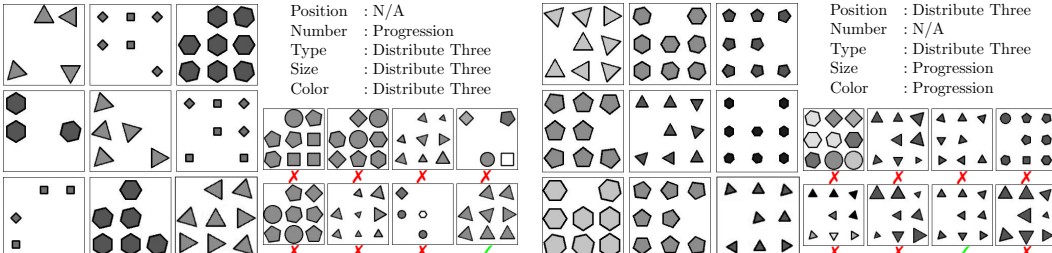

Figure 3: Examples of RPM instances with the missing entries filled by solutions generated by the ALANS$^2$ learner. Ground-truth relations are also listed. Note the generated results do not look exactly like the correct choices due to random rotations during rendering, but they are semantically correct.

## 5 CONCLUSION

In this work, we propose the ALgebra-Aware Neuro-Semi-Symbolic (ALANS$^2$) learner, echoing a normative theory in the connectionist-classicist debate that an algebraic treatment in a cognitive architecture should improve a model's systematic generalization ability. In experiments, we show that with such an algebraic treatment, the neuro-semi-symbolic learner achieves superior performance in three RPM domains reflective of systematic generalization.

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

# A INDUCING AND EXECUTING OPERATORS

In the main text, we examplify the induction and the execution process using a binary operator. Here, we discuss other details regarding the formulation for all three types of operators, *i.e.*, unary, binary, and ternary.

**Unary Operator** To induce the unary operator $\mathcal{T}_u^a$ for an attribute $a$, we solve the following optimization problem

$$\mathcal{T}_u^a = \arg\min_{\mathcal{T}} \ell_u^a(\mathcal{T}) = 1/5 \times \left( \mathbb{E}\left[ \left\| M(b_{o,1}^a)\mathcal{T} - M(b_{o,2}^a) \right\|_F^2 \right] + \mathbb{E}\left[ \left\| M(b_{o,2}^a)\mathcal{T} - M(b_{o,3}^a) \right\|_F^2 \right] + \right.$$
$$\mathbb{E}\left[ \left\| M(b_{o,4}^a)\mathcal{T} - M(b_{o,5}^a) \right\|_F^2 \right] + \mathbb{E}\left[ \left\| M(b_{o,5}^a)\mathcal{T} - M(b_{o,6}^a) \right\|_F^2 \right] +$$
$$\left. \mathbb{E}\left[ \left\| M(b_{o,7}^a)\mathcal{T} - M(b_{o,8}^a) \right\|_F^2 \right] \right) + \lambda_u^a \left\| \mathcal{T} \right\|_F^2 , \tag{S1}$$

where the indexing follows the row / column major. By taking the derivative with respect to $\mathcal{T}$ and setting it to be $\mathbf{0}$, we have the following solution,
$$\mathcal{T}_u^a = A^{-1}B \tag{S2}$$
where, assuming independence,
$$A = \mathbb{E}\left[ M(b_{o,1}^a)^T M(b_{o,1}^a) \right] + \mathbb{E}\left[ M(b_{o,2}^a)^T M(b_{o,2}^a) \right] + \mathbb{E}\left[ M(b_{o,4}^a)^T M(b_{o,4}^a) \right] +$$
$$\mathbb{E}\left[ M(b_{o,5}^a)^T M(b_{o,5}^a) \right] + \mathbb{E}\left[ M(b_{o,7}^a)^T M(b_{o,7}^a) \right] + 5\lambda_u^a I \tag{S3}$$
and
$$B = \mathbb{E}\left[ M(b_{o,1}^a)^T \right] \mathbb{E}\left[ M(b_{o,2}^a) \right] + \mathbb{E}\left[ M(b_{o,2}^a)^T \right] \mathbb{E}\left[ M(b_{o,3}^a) \right] + \mathbb{E}\left[ M(b_{o,4}^a)^T \right] \mathbb{E}\left[ M(b_{o,5}^a) \right] +$$
$$\mathbb{E}\left[ M(b_{o,5}^a)^T \right] \mathbb{E}\left[ M(b_{o,6}^a) \right] + \mathbb{E}\left[ M(b_{o,7}^a)^T \right] \mathbb{E}\left[ M(b_{o,8}^a) \right]. \tag{S4}$$

Note that as long as $\lambda_u^a > 0$, $A$ is a symmetric positive definite matrix and hence is invertible. Compared to the binary case, the unary operator can be regarded as a special binary operator where one of the operand is a constant, absorbed into operator learning, and jointly solved.

To predict the answer representation, we solve another optimization problem, *i.e.*,
$$\widehat{M_u^a} = \arg\min_{M} \ell_u^a(M) = \mathbb{E}\left[ \left\| M(b_{o,8}^a)\mathcal{T}_u^a - M \right\|_F^2 \right]. \tag{S5}$$
Taking its derivative and setting it to $\mathbf{0}$, we have
$$\widehat{M_u^a} = \mathbb{E}\left[ M(b_{o,8}^a) \right] \mathcal{T}_u^a. \tag{S6}$$
Note that this is exactly the execution of the learned operator.

**Binary Operator** The optimization problem for the binary case can be expanded as
$$\mathcal{T}_b^a = \arg\min_{\mathcal{T}} \ell_b^a(\mathcal{T}) = 1/2 \times \left( \mathbb{E}\left[ \left\| M(b_{o,1}^a)\mathcal{T}M(b_{o,2}^a) - M(b_{o,3}^a) \right\|_F^2 \right] + \right.$$
$$\left. \mathbb{E}\left[ \left\| M(b_{o,4}^a)\mathcal{T}M(b_{o,5}^a) - M(b_{o,6}^a) \right\|_F^2 \right] \right) + \lambda_b^a \left\| \mathcal{T} \right\|_F^2 . \tag{S7}$$
We note that, assuming independence, the solution satisfies
$$\mathbb{E}\left[ M(b_{o,1}^a)^T M(b_{o,1}^a) \right] \mathcal{T} \mathbb{E}\left[ M(b_{o,2}^a)M(b_{o,2}^a)^T \right] +$$
$$\mathbb{E}\left[ M(b_{o,4}^a)^T M(b_{o,4}^a) \right] \mathcal{T} \mathbb{E}\left[ M(b_{o,5}^a)M(b_{o,5}^a)^T \right] + 2\lambda_b^a \mathcal{T} \tag{S8}$$
$$= \mathbb{E}\left[ M(b_{o,1}^a)^T \right] \mathbb{E}\left[ M(b_{o,3}^a) \right] \mathbb{E}\left[ M(b_{o,2}^a)^T \right] + \mathbb{E}\left[ M(b_{o,4}^a)^T \right] \mathbb{E}\left[ M(b_{o,6}^a) \right] \mathbb{E}\left[ M(b_{o,5}^a)^T \right].$$
This is a linear matrix equation and can be turned into a linear equation by vectorization. Using $\text{vec}(A\mathcal{T}B) = A \otimes B \text{vec}(\mathcal{T})$ (Lancaster, 1970), where $\otimes$ denotes the Kronecker product, we have
$$\text{vec}(\mathcal{T}_b^a) = A^{-1}B, \tag{S9}$$
where
$$A = \mathbb{E}\left[ M(b_{o,1}^a)^T M(b_{o,1}^a) \right] \otimes \mathbb{E}\left[ M(b_{o,2}^a)M(b_{o,2}^a)^T \right] +$$
$$\mathbb{E}\left[ M(b_{o,4}^a)^T M(b_{o,4}^a) \right] \otimes \mathbb{E}\left[ M(b_{o,5}^a)M(b_{o,5}^a)^T \right] + 2\lambda_b^a I \tag{S10}$$
and
$$B = \text{vec}\left( \mathbb{E}\left[ M(b_{o,1}^a)^T \right] \mathbb{E}\left[ M(b_{o,3}^a) \right] \mathbb{E}\left[ M(b_{o,2}^a)^T \right] \right) +$$
$$\text{vec}\left( \mathbb{E}\left[ M(b_{o,4}^a)^T \right] \mathbb{E}\left[ M(b_{o,6}^a) \right] \mathbb{E}\left[ M(b_{o,5}^a)^T \right] \right). \tag{S11}$$
Note that $A$ is also symmetric positive definite given positive $\lambda_b^a$ and hence invertible.

The predicted answer representation is given by

$$\widehat{M_b^a} = \arg\min_M \ell_b^a(M) = \mathbb{E}\left[\left\|M(b_{o,7}^a)\mathcal{T}_b^a M(b_{o,8}^a) - M\right\|_F^2\right], \tag{S12}$$

which can be solved by executing the induced binary operator $\widehat{M_b^a} = \mathbb{E}\left[M(b_{o,7}^a)\right]\mathcal{T}_b^a\mathbb{E}\left[M(b_{o,8}^a)\right]$.

**Ternary Operator** A ternary operation can be regarded as an unary operation on elements defined on rows / columns. Specifically, we propose to construct the algebraic representation of a row / column by concatenating the algebraic representation of each panel in it, *i.e.*,

$$M(b_{o,i}^a, b_{o,i+1}^a, b_{o,i+2}^a) = [M(b_{o,i}^a); M(b_{o,i+1}^a); M(b_{o,i+2}^a)]. \tag{S13}$$

Then the ternary operator can be solved by

$$\mathcal{T}_t^a = \arg\min_{\mathcal{T}} \ell_t^a(\mathcal{T}) = \mathbb{E}\left[\left\|M(b_{o,1}^a, b_{o,2}^a, b_{o,3}^a)\mathcal{T} - M(b_{o,4}^a, b_{o,5}^a, b_{o,6}^a)\right\|_F^2\right] + \lambda_t^a\left\|\mathcal{T}\right\|_F^2. \tag{S14}$$

Similar to the unary case discussed above,

$$\mathcal{T}_t^a = A^{-1}B \tag{S15}$$

where

$$A = \mathbb{E}\left[M(b_{o,1}^a, b_{o,2}^a, b_{o,3}^a)^T M(b_{o,1}^a, b_{o,2}^a, b_{o,3}^a)\right] + \lambda_t^a I \tag{S16}$$

and

$$B = \mathbb{E}\left[M(b_{o,1}^a, b_{o,2}^a, b_{o,3}^a)^T\right]\mathbb{E}\left[M(b_{o,4}^a, b_{o,5}^a, b_{o,6}^a)\right]. \tag{S17}$$

Correspondingly, the answer representation can be obtained by first executing the ternary operator $\mathbb{E}\left[M(b_{o,4}^a, b_{o,5}^a, b_{o,6}^a)\right]\mathcal{T}_t^a$ and slicing it from the result.

To compute the operator distribution, we model it based on the fitness of each operator type,

$$P(\mathcal{T}^a = \mathcal{T}_u^a \mid \{I_{o,i}\}_{i=1}^8) \propto \exp(-\ell_u^a(\mathcal{T}_u^a)) \tag{S18}$$

$$P(\mathcal{T}^a = \mathcal{T}_b^a \mid \{I_{o,i}\}_{i=1}^8) \propto \exp(-\ell_b^a(\mathcal{T}_b^a)) \tag{S19}$$

$$P(\mathcal{T}^a = \mathcal{T}_t^a \mid \{I_{o,i}\}_{i=1}^8) \propto \exp(-\ell_t^a(\mathcal{T}_t^a)). \tag{S20}$$

# B  INSTANCES OF OPERATORS

In the original work of Zhang et al. (2019a) and Hu et al. (2020), there are four operators: Constant, Progression, Arithmetic, and Distribute of Three. Progression is parameterized by its step size ($\pm 1/2$). Arithmetic includes addition and subtraction. And Distribute of Three is implemented as shifting and can be either a left shift or a right one. Note that Constant can be regarded as special Progression with a step size of 0. In this work, we group all four operators into three types: unary (Constant and Progression), binary (Arithmetic), and ternary (Distribute of Three).

To study systematic generalization in abstract relation learning, we use the RPM generation method proposed in (Zhang et al., 2019a; Hu et al., 2020) and carefully split data into three regimes:

- Systematicity: The training set and the test set contain all three types of operators but disjoint instances. Specifically, the training set has Constant, Progression of $\pm 1$, addition in Arithmetic, and left shift in Distribute of Three, while in the test set there are Progression of $\pm 2$, subtraction in Arithmetic, and right shift in Distribute of Three.

- Productivity: The training set contains only unary operators and the test set only binary operators. Specifically, the training set has Constant and all instances of Progression, while the test set all instances of Arithmetic.

- Localism: The training set contains only binary operators and the test set only unary operators. Specifically, the training set has all instances of Arithmetic and the test set Constant and all instances of Progression.

Please see Figs. S1 to S3 for examples in the three splits.

# C  IMPLEMENTATION DETAILS

## C.1  NETWORK ARCHITECTURE

We use a LeNet-like architecture (LeCun et al., 1998) for each branch of the object CNN. See Table S1 for the design. Note that the object CNN consists of four branches, including objectiveness,

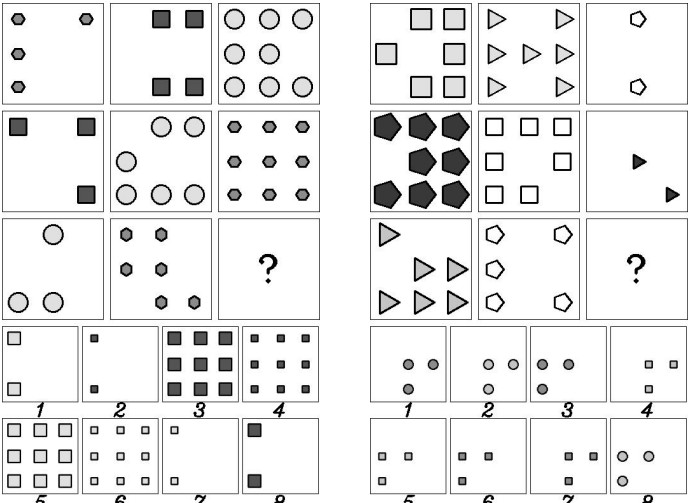

Figure S1: A training example (left) and a test example (right) in the systematicity split. Note that in the training example, the arithmetic relation (in number) is addition and the shifting is always a left shift (in type, size, and color). In the test example, the shifting becomes a right shift (in type), the size progression has a step of 2, and color arithmetic becomes subtraction.

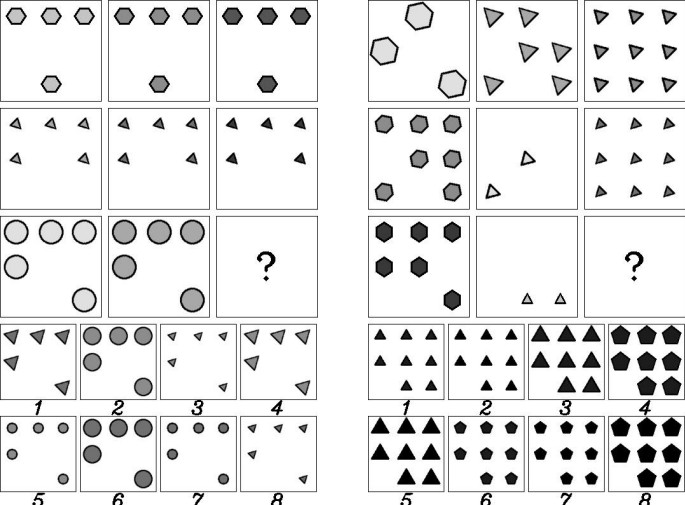

Figure S2: A training example (left) and a test example (right) in the productivity split. Note that in the training example, the constant rule is applied to the number, type, and size, while the progression rule is applied on color. In the testing example, the arithmetic rule is applied on all attributes.

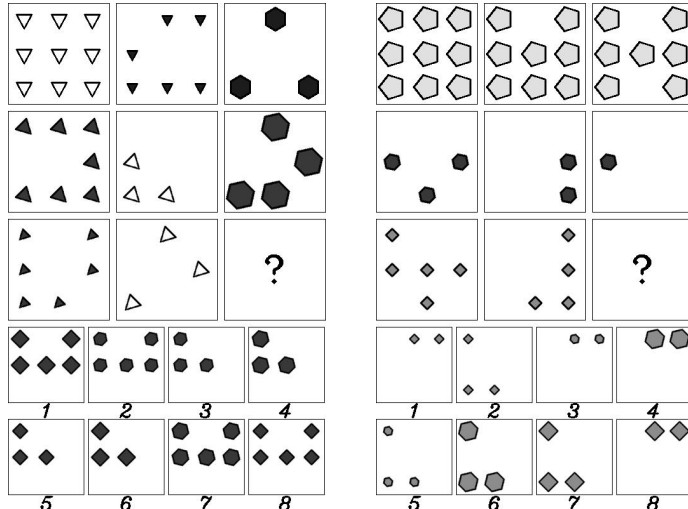

Figure S3: A training example (left) and a test example (right) in the localism split. Note that in the training example, the arithmetic rule is on all attributes. In the test example, the progression rule is applied on number and the constant rule on all other attributes.

Table S1: The network architecture used for each branch of the object CNN.

| Operator | Parameters |
|---|---|
| Convolution | $[6, 5, 1]$ |
| BatchNorm | 6 |
| SoftPlus | |
| MaxPool | 2 |
| Convolution | $[16, 5, 1]$ |
| BatchNorm | 16 |
| SoftPlus | |
| MaxPool | 2 |
| Linear | 120 |
| SoftPlus | |
| Linear | 84 |
| SoftPlus | |
| Linear | $m$ |
| LogSoftMax | |

type, size, and color. The parameters for Convolution denote the output channel size, kernel size, and stride, respectively. A BatchNorm layer is parameterized by the number of channels, whereas a MaxPool layer by its stride. An output size is used to specify a Linear layer's parameter. $m$ equals 2, 5, 6, 10 for objectiveness, type, size, and color, respectively. For numerical stability, we use LogSoftMax to turn a probability simplex into its log space.

## C.2 OTHER HYPERPARAMETERS

For the inner regularized linear regression, we set different regularization coefficients for different attributes but, for the same attribute, we keep them the same across all three types of operators. For position, $\lambda = 10^{-4}$. For number, $\lambda = 10^{-6}$. For type, $\lambda = 10^{-6}$. For size, $\lambda = 10^{-6}$. For color, $\lambda = 5 \times 10^{-7}$. All of the regularization terms in Eq. (9) in the main text are set to be 1 and $\{M_0^a\}$ and $\{M^a\}$ are initialized as $2 \times 2$ square matrices.

For training, we first train for 10 epochs parameters regarding objectiveness, including the objectiveness branch, and the representation matrices on position and number. We then perform 2 rounds of cyclic training on parameters regarding type, size, and color, each of which experiences 10 epochs of updates in a round. Finally, we fine-tune all parameters for another 10 epochs, totaling up to 80 training epochs. The entire system is optimized using ADAM (Kingma & Ba, 2014) with a learning rate of $9.5 \times 10^{-5}$.

## D  MARGINALIZATION FOR OTHER ATTRIBUTES

For the attribute of position, we denote its value as $R^o$, a binary vector of length $N$, with each entry corresponding to one of the $N$ windows. Then

$$P(\text{Position} = R^o) = \prod_{j=1}^{N} P(r_j^o = R_j^o), \tag{S21}$$

where $P(r_j^o)$ denotes the $j$th region's estimated objectiveness distribution returned by a CNN as in the main text.

For the attribute of type, the panel attribute of type being $k$ is evaluated as

$$P(\text{Type} = k) = \sum_{R^o} \left( \prod_{j, R_j^o=1} P(r_j^t = k) \right) P(\text{Position} = R^o), \tag{S22}$$

where $P(r_j^t)$ denotes the $j$th region's estimated type distribution returned by a CNN.

The computation for size and color is exactly the same as type, except that we use the region's estimated size and color distribution returned by a CNN.

## E  RELATED WORK ON NEURAL THEOREM PROVING

Combining neural architectures with symbolic reasoning has a long history in the field of theorem proving (Garcez et al., 2012), with early works dated back to propositional rules (Shavlik & Towell, 1991; Towell & Shavlik, 1994; Garcez & Zaverucha, 1999). Later works extend the propositional rules to first-order inference (Shastri, 1992; Ding, 1995; França et al., 2014; Sourek et al., 2015; Cohen, 2016). More recent works include the Logic Tensor Networks (Serafini & Garcez, 2016) and the NTP model (Rocktäschel & Riedel, 2017). The former grounds first-order logics and supports function terms, while the latter is constructed from Prolog's backward chaining and is related to Komendantskaya (2011); Hiolldobler (1990) but supports function-free terms. DeepProbLog (Manhaeve et al., 2018) further improves on NTP by focusing on tight interactions between a neural component and subsymbolic representation and parameter learning for both the neural and the logic components. Evans & Grefenstette (2018) introduces a differentiable rule induction process, though not integrating the neural and symbolic components. Our work is related to the stream of work on neural theorem proving. However, we formulate the relation induction process as continuous optimization rather than logical induction.

## F  MORE ON NEURAL VISUAL PERCEPTION

- **Why not train a CNN to predict the position and number of objects?** The CNN is trained to predict the type, size, color, and object existence in a window. The object existence in windows is marginalized to be a Number distribution and Position distribution. This is a light-weight method for object detection. Nevertheless, it is also possible to use a Fast-RCNN like method to predict object positions (this implies number) directly. However, in this way, the framework loses the probabilistic interpretation (the object proposal branch is currently still deterministic), and we cannot perform end-to-end learning.

- **How does the CNN predict the presence of an object, its type, size, and color given that it is not trained to do that?** For each window, the CNN outputs 4 softmaxed vectors, corresponding to the probability distributions of object existence, object type, object size, and object color. The spaces for these attributes are pre-defined. CNN's weights are then jointly trained in the framework. Such a design follows recent neuro-symbolic methods (Mao et al., 2019; Han et al., 2019) that also rely on the implicitly trained representation. In short, we assign semantics to the implicitly trained representation (probability distributions for attributes), performs marginalization and reasoning **as if** they are ground-truth attribute distributions, and jointly train using only the problem's target label.

