# OpenReview forum: "Learning Algebraic Representation for Abstract Spatial-Temporal Reasoning"
_ICLR.cc/2021/Conference — Reject_

### Official Review · AnonReviewer3 · 2020-10-23
**Good but could benefit further clarification and discussion on the applications of this work**

**Rating:** 6
**Confidence:** 3

**Review:**

**Summary**.
This work proposes a new learner bridging the gap between connectionists and classicists in the task of Raven’s Progressive Matrices (RPM). It relies on a CNN to extract visual features and then uses an algebraic abstract reasoning module to infer the operators of an RPM instance, which allows applying the inferred operator on the RPM instance to predict potential solutions according to various attributes. The most likely solution according to the ensemble of the attributes is then selected as an answer.

**Pros**.
- This work is significant as neural models still struggle in systematically generalizing in reasoning tasks.
- The general idea of the paper is easy to follow but that does not mean the proposed method is trivial, far from it.
- Experiments are clearly described and the authors give particular attention to split their dataset to systematically test for generalization.

**Cons**.
- Additional clarifications and motivations could benefit the paper (see **Questions & Suggestions** below).
- While yielding impressive results and targeting a fundamental issue of neural models, it is not clear how the proposed technique could be applied to other domains. As of now the paper focus only on the RPM task. It would be beneficial to open up the discussion to potential applications. It would be nice to know what the authors think are the potential applications of their learner. How would it be applied to other reasoning tasks?

**Questions & Suggestions**.
- The CNN is used to predict the presence of an object, its type, its size, and its color while a belief inference engine is used to predict the position and number of objects. It is not clear why the CNN was not also used to predict the position and number of objects, thus questioning the motivation behind the belief inference engine.
Would training a CNN to predict the position and number of objects also help?
- It is not clear how the CNN predicts the presence of an object, its type, size, and color given that it is not trained to do that. Do you produce 4 feature vectors, apply a softmax to each, and arbitrarily decide which feature vector will predict each object attribute? If so the results in Table2 are very impressive and hard to believe. How do you decide which feature vector will predict each object attribute?
- Suggestion: as an ablation study, it would be nice to also compare the performance of ALANSS with randomly initialized and fixed CNNs. This would give an estimate on the advantage of using CNNs as neural modules. One could hypothesize that the robustness of the method comes mostly from the symbolic abstraction rather than the neural representation.
- Neural Theorem Proving also tries to bridge the gap between connectionists and classicist approaches. Can it be related to this work? It would be nice to discuss this in the Related Work section.

---

> ### Author Response · Authors · 2020-11-12
> **Response to R3**
>
> Thank you for your support and appreciation in our work to improve systematic generalization in the visual reasoning task.
>
> It is an excellent question on the broader application of the proposed algebraic representation and the optimization-as-reasoning method---it is also a question we ask ourselves. However, the RPM problem is so unique in few-shot induction and logic numerical reasoning that we have not found a more practical vision application to apply; designing alternative or better problems has been challenging in cognitive science for decades. On this point, any advice on a broader application is welcome. And we are happy to report the performance if the limited time permits.
>
> For your questions:
>
> * The CNN is trained to predict the type, size, color, and *object existence in a window*. The object existence in windows is marginalized to be a Number distribution and Position distribution. This is a light-weight method for object detection. Nevertheless, it is also possible to use a Fast-RCNN like method to predict object positions (this implies number) directly. However, in this way, the framework loses the probabilistic interpretation (the object proposal branch is currently still deterministic), and we cannot perform end-to-end learning.
>
> * For each window, the CNN outputs 4 softmaxed vectors, corresponding to the probability distributions of object existence, object type, object size, and object color. The spaces for these attributes are pre-defined. CNN's weights are then jointly trained in the framework. Such a design follows recent neuro-symbolic methods [1, 2] that also rely on the implicitly trained representation. In short, we assign semantics to the implicitly trained representation (probability distributions for attributes), performs marginalization and reasoning as if they are ground-truth attribute distributions, and jointly train using only the problem's target label.
>
> * In fact, the visual representation is essential. Keeping the visual CNN as randomly initialized and training only the symbolic part gives a chance-level performance of around 12.5%. This is actually expected: if your vision cannot correctly tell you what you are looking at, there are minimum chances that you get the problems right.
>
> * We will update the related work section to include a discussion on Neural Theorem Proving if accepted. For now, please refer to a new section in appendix for a draft version of it, and if some essential references are missing, we will include them after another revision.
>
> [1] Mao et al. The neurosymbolic concept learner: Interpreting scenes, words, and sentences from natural supervision. ICLR 2019.
>
> [2] Han et al. Visual concept-metaconcept learning. NeurIPS 2019.
>
> If there is still anything unclear, please let us know and we are more than happy to discuss.

---

> > ### Comment · AnonReviewer3 · 2020-11-23
> > **Final comment**
> >
> > Thank you for your reply and explanations. It makes more sense now.
> >
> > I think the paper would benefit from the two explanations you gave in your first 2 points, namely:
> > - explaining why you can't train the CNN to predict more attributes. That point is still not 100% clear to me.
> > - saying that "the CNN outputs 4 softmaxed vectors" and that "The spaces for these attributes are pre-defined." and that you "performs marginalization and reasoning **as if** they are ground-truth attribute distributions".
> >
> > Eventually, I think that saying "the RPM problem is so unique in few-shot induction and logic numerical reasoning that we have not found a more practical vision application to apply" is a potential weakness. I would suggest the authors to mention a few motivating points why it is useful to train neural models to solve the RPM task. Maybe it shares similar challenges as other real-world problems and working with RPM makes progress more tractable?

---

> > > ### Author Response · Authors · 2020-11-23
> > > **Reply to R3**
> > >
> > > Thank you for your reply.
> > >
> > > We'll include the two points in another revision of the work. We also hope that the reviewer can clarify on which point he/she is not clear about, so that we can do our best to clarify.
> > >
> > > In a theoretic point of view, the task of RPM was initially proposed as a challenge on few-shot abstract reasoning. The task is meaningful at least in the following aspects:
> > >
> > > * The problem of inducing the hidden relations from a few examples is, in itself, an interesting problem. Human performance on this abstract reasoning task has been found to be correlated with *general intelligence* (the *g* factor) and *fluid intelligence* (the ability to quickly reason with information to solve new, unfamiliar problems, independent of prior knowledge) [5]. Therefore, improvement on performance on this task may potentially lead to improvement on general AI.
> > > * Early works found out [3, 4] that Amazonians, absent of schooling, could still correctly answer a nontrivial number of RPM-style problems. How do they transfer knowledge from their living experience into the relational learning problem?
> > > * We quote from the recent [**AAAI Fall Symposium on Conceptual Abstraction and Analogy in Natural and Artificial Intelligence**](https://sites.google.com/pdx.edu/abstractionfall2020) (click to see details) to support the meaningfulness of this task: "Understanding what concepts are—how they are formed, can be abstracted and flexibly used in diverse situations via analogy, how they compose to produce new concepts—is not only key to a deeper understanding of intelligence, but will be essential for engineering non-brittle AI systems, ones that can robustly adapt their knowledge to diverse situations and modalities."
> > >
> > > But we admit that solving abstract reasoning problems in a human-like way could be only a quest for general intelligence and, at the current stage, may not have found a practical application in the real world: the same question is proposed in the Symposium mentioned above and unfortunately not resolved yet. However, we are optimistic that some techniques developed for solving the task may be adopted for other reasoning problems in the future.
> > >
> > > [3] Dehaene, Stanislas, et al. "Core knowledge of geometry in an Amazonian indigene group." Science 311.5759 (2006): 381-384.
> > > [4] Izard, Véronique, et al. "Flexible intuitions of Euclidean geometry in an Amazonian indigene group." Proceedings of the National Academy of Sciences 108.24 (2011): 9782-9787.
> > > [5] Hofstadter, Douglas R. Fluid concepts and creative analogies: Computer models of the fundamental mechanisms of thought. Basic books, 1995.

---

### Official Review · AnonReviewer1 · 2020-10-28
**Learning Algebraic Representation for Abstract Spatial-Temporal Reasoning**

**Rating:** 7
**Confidence:** 1

**Review:**

The author(s) propose an architecture ALANS^2, primarily focused on the task
of Raven's Progressive Matrices (RPM, popularly known as IQ tests).
To solve it, they join an image classifier on the individual fields
of the 3x3 picture matrix with a "reasoning backend". In the backend,
they model the high level rules of the task by matrix multiplication.
Given a problem instance, they find a such an operator (matrix) that fits
the first two rows of the instance the best, and use this operator
to determine what picture belongs to the missing field.

For training, they use not only the knowledge of the correct answer but also
the correct operator which should be found (the "rule" which should be discovered).
Experimental results show that they exceed by a margin
the previous models used for this task.

The task is interesting, in fact some research has been done on trying to
learn intelligence tests in various ways and a number of the relevant ones
are properly cited in the paper.

The results presented show an improvement with respect to previous
results that I find enough to deserve publication.

The article is well written apart from the typo mentioned below.

Typo:
Pg3, part "Abstract Visual Reasoning" last sentence: fronend -> frontend

---

> ### Author Response · Authors · 2020-11-12
> **Response to R1**
>
> Thank you for your time and efforts in reviewing our work, and most importantly, thanks for the support.
>
> We corrected the typo in the latest revision and will include any essential references you believe are missing.

---

### Official Review · AnonReviewer4 · 2020-10-28
**Paper 950 Review**

**Rating:** 5
**Confidence:** 4

**Review:**

This paper proposed ALANS, a semi neuro-symbolic learner specifically designed for improved systematic generalization on RAVEN dataset.

Pros:
1. The proposed model integrates learnable symbolic operators that are similar to the ones used in Neural Theorem Proving. The operators are shown to have improved systematic generalization performance.
2. Introduced a new generalization testing dataset based on RAVEN dataset.

Cons:
1. The proposed model relies on strong assumptions specific to RPM, and even RAVEN dataset. For example, In Equation 3, the authors assume the knowledge that relations can only exist in rows of the RAVEN dataset. However for PGM dataset (another well-known RPM dataset), the relations can exist either in rows or columns, or both. I hypothesize that training operators when there is no guaranteed existence of relations will lead to noisy gradients. I have doubts that the proposed method can be readily adapted to PGM, not to mention other types of reasoning tasks.
2. While the proposed model is shown to perform well on the dataset generated based on RAVEN, I think the authors should show results on PGM dataset as well, considering that PGM already have a few data splits to test systematic generalization.

Clarity:
In general the paper is written in a clear and understandable manner. The only part that confuses me is about the perception CNN module. It is not clear to me how the model learns to disentangle object attributes such as color and position? Is there any auxiliary labels to help induce such disentanglement, or does the model learn to disentangle the attributes because of the way authors marginalize the object attributes in a diagram? The authors only show the marginalization for the attribute 'number'. I suggest the authors to include the marginalization for all other attributes in the Appendix for better understandability.

Summary:
Overall I think this paper propose an interesting approach that improves systematic generalization on RAVEN datasets. But I have doubts on its adaptabilities to PGM dataset and other types of reasoning tasks. And there is also some lacking details that hinders understanding of the paper. I will raise my score if the authors can (1) show improved performance on generalization data splits on PGM dataset, and (2) clarify about how the model learns to disentanglement the object attributes.

---

> ### Author Response · Authors · 2020-11-12
> **Response to R4**
>
> Thank you for your time in reviewing our work and pointing out our potential issues. Thanks for your advice.
>
> For your concerns:
>
> * Column and row: adding the relations in columns is not the primary problem; we can compare the best rule column-wise and the best rule row-wise and measure the probability as done in Eq (4). Nevertheless, adding this one more step will indeed add noise in gradients.
>
> * PGM and PGM splits: the main point in this work is not to claim superiority in the traditional I.I.D. learning setting but how the specific algebraic inductive bias can improve neural methods in O.O.D. settings in *systematic generalization for different relations*. The PGM dataset's splits focus on generalization in visual attributes and *not on the systematic generalization in relational reasoning*. The visual generalization part is essential, and a more generalizable visual perception part is undoubtedly desirable, but it is not the point of this work.
>
> * CNN disentanglement: the CNN predicts for each window the object existence, object type, object size, and object color. The object existence in windows is marginalized to be a Number distribution and Position distribution. This is a light-weight method for object detection. However, it is also possible to use a Fast-RCNN like method to predict object positions (this implies number) directly. We choose the light-weight method and jointly train the CNN in the entire framework, *without object attributes*. The disentanglement is achieved in a way similar to [1, 2]; in short, we assign semantics to the hidden representation (probability distributions for attributes), performs marginalization and reasoning as if they are ground-truth attribute distributions, and jointly train using only the problem's target label. We show the marginalization process for other attributes in a new section in the appendix.
>
> * Neural Theorem Proving: We included a draft version for related work in neural theorem proving in a new section in the appendix; we welcome any comments on this part.
>
> [1] Mao et al. The neurosymbolic concept learner: Interpreting scenes, words, and sentences from natural supervision. ICLR 2019.
>
> [2] Han et al. Visual concept-metaconcept learning. NeurIPS 2019.
>
> If there is still anything unclear, please let us know and we are more than happy to discuss.

---

> > ### Comment · AnonReviewer4 · 2020-11-23
> > **About Systematic Generlization**
> >
> > Thank you for your reply. Could you explain a bit more why the PGM dataset's splits is out of the scope? Your definition of systemacity is:
> >
> > Systematicity: the training set contains only a subset of instances for each type of relation, while
> > the test set all other relation instances.
> >
> > I am not sure why certain splits in PGM are not testing this. For example the training set of extrapolation split contains only objects with attributes lower than a certain threshold for each type of relation, while the test set contains objects with attributes higher than the threshold. Can you explain the differences? Thanks.

---

> > > ### Author Response · Authors · 2020-11-23
> > > **Reply to R4**
> > >
> > > Thank you for the reply.
> > >
> > > Our work tests models' generalization in *relation learning* rather than *attribute learning*. For example, the extrapolation split focuses on how a model trained on the lower half of an attribute can performs on the upper half.  But in this paper, we focus on how relation learning can be generalized. Say, how learning of the relation of **+1** enables the understanding of **+2**.
> > >
> > > We do not use PGM because
> > >
> > > * PGM's rule set does not support such generalization tests: we cannot control systematicity (variations) in a rule and there are not rule pairs that support productivity (recursion) and localism (converse of recursion).
> > > * PGM's generalization focus is not on *relations*, but more on *across attributes*. And the splits do no follow the principles of *systematic generalization*.
> > >
> > > That's the reason why we believe "The visual generalization part is essential, and a more generalizable visual perception part is undoubtedly desirable, but it is not the point of this work."

---

### Official Review · AnonReviewer2 · 2020-11-01
**initial review**

**Rating:** 5
**Confidence:** 4

**Review:**

Initial review (2020.11.01)

Review:
This paper addresses a specialized solution to combine neural networks and linear models for Raven progressive matrices (RPM). The solver is composed of convolutional neural networks (CNN) as an image perception module and a reasoning module with linearly parameterized operators to represent successive operations in the RPM. The operators are trained with regularized linear regressors, which are lightweight capable to adapt on-the-fly for each instance. The overall architecture enables end-to-end training and can be used for generation. For performance evaluation on two kinds of automatically generated problem sets, the authors test the proposed method with respect to 3 extrapolatory settings such as systematicity, productivity, and localism, which seems to be newly defined tasks for RPM. Their method outperforms pure neural state-of-the-art methods with a large margin over all of settings.
The proposed methods solve RPMs well with available components such as CNNs and linear models, which are able to be tuned on-the-fly. Also, this paper provides systematic results to easily comprehend the role of modules with new settings (systematicity, productivity, and localism) for RPMs. On the other hands, my major concerns are two-fold: (1) one of important research questions introduced in this work, “what constitutes such an algebraic inductive bias?” seems not so clearly answered in this paper. (2) the comparative result in the paper just considers only pure connectionist methods without considering other (semi-)symbolic approaches. Since this work utilizes additional problem information (e.g., specified features) for modeling than compared neural methods, I think it would be better to justify the position of this work by comparing various approaches including search-based solver, symbolic, and the neural methods to figure out the superiority of this approach.
As a result, I vote for marginally below acceptance threshold.

Pros:
-	The authors provide a specialized solution for RPM well with CNNs and linear models.
-	They introduce new 3 settings of RPMs and report systematic experimental results.

Concerns:
-	In the comparative study, the authors report only the systematicity, the productivity, and the localism, not the "accuracy" used in the cited related works. Due to the lack of the information, the readers can not directly compare them with respect to the previous viewpoint.
-	If the result of other various approaches for RPM not only pure connectionist methods but also search-based solver, symbolic, and neuro-symbolic approaches, it would be easier to find the niche of this work and its superiority.
-	As I mentioned above, the research questions, “what constitutes such an algebraic inductive bias?” seems not so clearly answered in this paper. Could you discuss this issue with the result in the paper?

Minors:
-	It would be better understandable to show examples for 3 settings of systematicity, productivity and localism on RPMs.
-	fare -> far?

---

> ### Author Response · Authors · 2020-11-12
> **Response to R2**
>
> Thanks for your time in reviewing our work and pointing out our potential issues. Thanks for your advice.
>
> For your concerns:
>
> * Algebraic inductive bias: our answer to the algebraic inductive bias is the Peano-Axiom-based encodings together with the representation-theory-based linear operations. Such representation supports easy encode and decode and is potentially more interpretable than a general feature map. The matrix product also enables easy optimization under our current formulation. It is also possible to formulate a more complex optimization problem under the representation, such as a general convex optimization. And very luckily, the gradients can still be backpropagated through using the implicit function theorem. From this perspective, we propose a prototype of the Peano-Axiom-based encodings and the representation-theory-based linear operations, and we believe there is room for future extension.
>
> * Symbolic methods: we have run a search-based symbolic method used for benchmarking in [1]. The method assumes knowledge for the hidden rules and perfect perception, and searches for the most appropriate rule when it encounters a new problem. As reported in [1], the method achieves 100% accuracy. If the reviewer believes there are any other search-based or symbolic methods that need comparing, we will do so if time permits. From this perspective, our work can be positioned as: we need to learn to perceive (there is perception uncertainty), and the hidden rule is to be induced on-the-fly based on constraints realized from vision, rather than given.
>
> * Accuracy: the metrics for each of the split is indeed accuracy; it is the proportion of the problems that are answered correctly. The main point in this work is not to claim superiority in the traditional I.I.D. learning setting but how the specific algebraic inductive bias can improve neural methods in O.O.D. settings in systematic generalization.
>
> * Examples: we have included a few examples for each of the split in a new section of the appendix. However, the differences actually lie in the distribution of relations; the visual part is the same.
>
> [1] Zhang et al. Raven: A dataset for relational and analogical visual reasoning. CVPR 2019.
>
> If there is still anything unclear, please let us know and we are more than happy to discuss.

---

### Decision · Program_Chairs · 2021-01-07
**Final Decision**

**Decision:**

Reject

**Comment:**

This paper was reviewed by 4 experts in the field. The reviewers raised their concerns on lack of novelty, unconvincing experiment, and the presentation of this paper, While the paper clearly has merit, the decision is not to recommend acceptance. The authors are encouraged to consider the reviewers' comments when revising the paper for submission elsewhere.